

# Relative dispersion and kinematic properties of the coastal submesoscale circulation in the southeastern Ligurian Sea

Pierre-Marie Poulain[1], Luca Centurioni[2], Carlo Brandini[3], Stefano Taddei[3], Maristella Berta[4], Milena Menna[5]

[1]Centre for Maritime Research and Experimentation (CMRE), La Spezia, 19126, Italy
[2]Scripps Institution of Oceanography, La Jolla, 92093, California
[3]Laboratorio di Monitoraggio e Modellistica Ambientale per lo sviluppo sostenibile (LAMMA), CNR, 50019, Sesto Fiorentino, Italy
[4]Istituto di Scienze Marine (ISMAR), CNR, La Spezia,19032,  Italy
[5]Istituto Nazionale di Oceanografia e di Geofisica Sperimentale (OGS), Sgonico (Trieste), 34010, Italy

*Correspondence to*: Pierre-Marie Poulain (pierre-marie.poulain@cmre.nato.int)

**Abstract.** An array of Lagrangian instruments (more than 100 drifters and a profiling float) was deployed for several days in the coastal waters of the southeastern Ligurian Sea to characterize the near-surface circulation at the submesoscale (< 10 km).  The drifters were trapped in an offshore-flowing filament and a cyclonic eddy that developed at the southwestern extremity of the filament. Drifter velocities are used to estimate differential kinematic properties (DKPs) and the relative dispersion of the near-surface

currents on scales as small as 100 m. The maximum drifter speed is ~50 cm/s. The DKPs within the cluster exhibit considerable spatial and temporal variability, with absolute values reaching the order of magnitude of the local inertial frequency. Vorticity prevails in the core of the cyclonic eddy, while strain is dominant at the outer edge of the eddy. Significant convergence was also found in the southwestern flow of the filament. The initial relative dispersion on small scales (100-200 m) is directly related to

some of the DKPs (e.g., divergence, strain and instantaneous rate of separation): The mean squared separation distance (MSSD) grows exponentially with time and the finite-size Lyapunov exponent (FSLE) is independent of scale. After 5-10 h of drift or for initial separations greater than 500 m, the MSSD and FSLE show smaller relative dispersion that decreases slightly with scale.

## 1 Introduction

Coastal filaments and eddies play an important role for the transport between coastal and deep open-sea waters, and are therefore critical to the local ecosystem dynamics and fisheries. They can be quite small (< 10 km, hereafter referred to as submesoscale) and evolve rapidly at daily or smaller timescales. They can be seen in satellite imagery of coastal areas, especially where rivers discharge water with different physical (e.g., temperature) or biological (e.g., chlorophyll or dissolved organic matter) properties into

the sea. Examples of coastal filaments and eddies detected by satellite imagery of sea surface temperature or chlorophyll concentration and observed by in-situ measurements in the oceans and semi-enclosed seas



can be found in numerous publications (e.g., Flament et al., 1985; Wong et al., 1988; Zatsepin et al., 2003; Poulain et al., 2004, 2020; Schroeder et al., 2011,2012; Schaeffer et al., 2017).

In-situ observations of coastal dynamics using traditional methods based on surveys with research vessels and moored instruments are not ideal for sampling high-frequency and small-scale dynamics, especially when there are hazards or limitations due to local fisheries and other coastal maritime activities. An alternative approach is to use numerous, low-cost, freely-drifting (Lagrangian) instruments deployed rapidly in a specific area and tracked over time (e.g., Mahadevan et al., 2017; D'Asaro et al., 2018). One such a sampling strategy was adopted off Livorno (Italy) in the southeastern Ligurian Sea (SLS; Figure 1) in October 2020 to provide three-dimensional (3D) spatial characterization and rapid temporal monitoring of the coastal environment at scales as small as ~100 m (Poulain, 2020).

Circulation in the SLS is dominated by the East Corsica Current (ECC), which flows northward between the islands of Corsica and Elba (Figure 1). The ECC varies seasonally (Astraldi and Gasparini, 1992) and is also characterized by velocity fluctuations with periods of 2-15 days with intermittent reversals (Astraldi et al., 1990). The ECC generally rotates clockwise around the island of Capraia, forming an anticyclonic eddy centred on the island (Poulain et al., 2012; Ciuffardi et al., 2016). This Ligurian or Capraia Eddy, is dominant in summer when the ECC is weak. Coastal circulation and dispersion in the SLS region have been described using ocean colour satellite imagery and drifter data (Schroeder et al., 2012; Poulain et al., 2020). Coastal currents were shown to vary strongly with local winds, including intermittent complete reversals in direction. Coastal dispersion was found to be an order of magnitude larger than in the offshore Ligurian Sea, and was significantly underestimated by numerical ocean circulation simulations (Schroeder et al., 2012). The objective of this work is to describe the spatial structure and temporal evolution of a particular submesoscale offshore-flowing filament and a small cyclonic eddy sampled by Lagrangian instruments in the coastal SLS, focusing on the local surface dispersion and the kinematic properties of the currents.

The experimental site was chosen east of the ECC and Ligurian Eddy about 15 km from the Italian coast (Figure 1). A cloud-free Moderate Resolution Imaging Spectro-radiometer (MODIS) chlorophyll concentration image taken on 8 October 2020 reveals several coastal filaments and eddies transporting nutrient-rich water offshore from the Italian coast. In particular, a filament extending tens of km in the southwest direction prevails near the northwestern edge of the drifter deployment array. On the same day, operational numerical simulations provided by the Copernicus Marine Environment Monitoring Service (CMEMS) at 1/24th degree (~4 km) horizontal resolution show a well-defined coastal area with fresher water to the East and North of our experimental site, mainly due to the outflow of the Arno River near Livorno. CMEMS currents are rather weak (< 10 cm/s) in this coastal area. In contrast, a noteworthy meandering ECC and Ligurian Eddy with speed reaching ~50 cm/s dominate the near-surface circulation offshore (Figure 1).

More than one hundred drifting instruments deployed quickly in a small array on the morning of 8 October 2020 were used to study the near-surface relative dispersion and kinematic properties of an offshore-flowing filament and cyclonic eddy. Additional drifters and float were deployed to provide



ancillary data on surface waves and vertical profiles of temperature, salinity and currents. All the drifting instruments deployed during the experiment are briefly described in Section 2, including information on their deployments and the processing of their data. Data analysis methods are also described. Results are presented and discussed in Section 3, focusing on the kinematic properties of the near-surface circulation and lateral relative dispersion. The results are discussed and conclusions are drawn in Section 4.

## 2. Data and Methods

### 2.1 Lagrangian instruments

The drifters and profiling float used in the coastal SLS are described in detail in Poulain (2020). Only a summary is provided below. Most drifters were Coastal Ocean Dynamics Experiment (CODE; Davis, 1985), Consortium for Advanced Research on Transport of Hydrocarbon in the Environment (CARTHE;

Novelli et al., 2017) and Palo Alto Research Center (PARC; Waterston et al., 2019; Cocker et al., 2022) drifters using GlobalStar or Iridium satellite telemetry systems. Global Positioning System (GPS) positions were measured every 5 to 20 minutes. They measured surface currents within 1 m of the sea surface. The effects of wind and waves on the motion of CODE and CARTHE drifters are comparable (Poulain et al., 2022). The main error is a wind-induced slip of about 0.1% of the wind speed (Poulain

and Gerin, 2019). The wind- and wave-induced slip of the PARC drifters has not yet been studied. A total of 50 CODE, 20 CARTHE (Berta et al., 2021), and 30 PARC drifters was deployed.

Additional Lagrangian instruments included: 1) the RIVER drifter, a CODE-like drifter equipped with a down-looking acoustic Doppler current profiler (ADCP) to measure relative current profiles between 2 and 20 m depth with a vertical resolution of 1 m; 2) the Surface Velocity Program (SVP) drifter (Niiler,

2001) with a drogue centered at 15–m nominal depth; 3) the Directional Wave Spectra (DWS) drifter (Centurioni et al., 2017) to measure the directional statistical properties of the surface wave; and 4) the Arvor-C float (André et al., 2010) to measure temperature and salinity profiles with a pumped conductivity, temperature and depth (CTD) sensor between the surface and ~120-m depth with 1 m vertical resolution. Five SVP, two RIVER, and three DWS drifters were operated.

The GPS position data of the drifters were quality controlled and interpolated at 0.5 h intervals using a kriging technique (Menna et al., 2017, and references therein). Velocities were calculated by finite differencing the interpolated positions (central difference with hourly interval).

### 2.2 Remotely sensed data and operational products

MODIS satellite images of chlorophyll concentration of the study area were used to describe the spatial

structure and temporal evolution of the surface circulation assuming that chlorophyll is a passive tracer advected by the surface horizontal currents. As previously shown in Poulain et al. (2020), chlorophyll concentration images were preferred over sea surface temperature images as they better represent circulation features. Since we are in a coastal area where a river drains nutrient-rich water, there is a



sharp contrast between coastal and offshore water, with the former being richer (higher chlorophyll) and
more turbid. Daily images have a horizontal resolution of 1 km.

Atmospheric data (wind speed and direction 10 m above sea level) and surface wave data (significant
wave height, main wave period and direction, Stokes drift) of the fifth generation ECMWF reanalysis
(ERA5) for the global climate and weather were downloaded from the Copernicus Climate Data Store
for October 2020 in the SLS. They are provided with a horizontal resolution of 0.25° (wind) and 0.5°
(waves).

### 2.3 Deployment strategy

In-situ data were collected as part of the Drifter Demonstration and Research 2020 (DDR20) experiment
(Poulain, 2020), which took place off the coast of Italy on 8-10 October 2020. DDR20 was a Rapid
Environment assessment (REA) exercise, whose general objective was the 3D characterization of the
oceanographic and acoustic environment using a network of compact and low-cost freely-drifting
instruments during a few days.  A total of 110 drifters and 1 float was quickly deployed in a 6x6 km$^2$
array in the coastal LSL (Figure 1) using two ships between 08:09 and 12:28 UTC on 8 October 2020.
The minimum distance between drifters at release was 0.5 km, if the drifters deployed at the same
time/position are not considered (Poulain, 2020). One third of these drifters and the float were
successfully recovered after about 2 days, starting at 09:22 UTC on 10 October 2020.

The experiment took place after a storm with westerly winds and waves up to 15 m/s and 2.5 m,
respectively, on 7 October (Figure 2). During the two days of drifter operations mentioned above, calm
meteorological conditions prevailed with winds less than 5 m/s and waves less than 0.5 m significant
wave height. The surface Stokes drift estimated by ERA5 was as large as 20 cm/s on 7 October, but
decreased to a few cm/s on subsequent days. Note that ERA5 underestimates the significant wave height
by up to 0.5 m compared to the DWS drifter measurements (Figure 2).

Unfortunately, 28 CODE drifters experienced transmission problems and did not transmit on 8 October
between 14:00 and 22:00 UTC (~8 h data gap) and between 9 October 10:00 UTC and 10 October 03:00
UTC (~17 h data gap). The interpolated drifter data were not used to estimate kinematic properties during
these gaps because of low horizontal resolution. Since the winds, waves and Stokes drift were relatively
weak during the experiment, all CODE, CARTHE and PARC drifters were combined to investigate the
kinematics and dispersion of the near-surface currents.

### 2.4 Analysis methods

The relative dispersion of a drifter cluster can be quantified using both the mean squared separation
distance (MSSD) as a function of time after deployment, $D^2(t)$, and the scale-dependent finite-size
Lyapunov exponent (FSLE) (Lacorata et al., 2001; Schroeder et al., 2011,2012; Corrado et al., 2017).
The MSSD of drifter pairs is defined as

$$D^2(t) = <|x^{(1)}(t)-x^{(2)}(t)|^2>, \tag{1}$$






where the superscripts denote the two drifters of the pair, that are located at position $x(t)$ at time $t$, and the brackets denote the average over all pairs with the same initial spacing. The FSLE, $\lambda$, is inversely proportional to the average time, $\tau$, for two drifters initially separated by $\delta_o$ to reach a prescribed separation, $\delta_f$:


$$\lambda(\delta_o, \delta_f) = 1/\tau \ln(\delta_f/\delta_o). \tag{2}$$

Following Schroeder et al. (2011) we chose an amplification factor $\delta_f/\delta_o = 1.2$. Relative dispersion by two-dimensional geophysical turbulence has the following dispersion regimes: exponential ($D^2 \sim e^{\alpha t}$, $\lambda =$ constant), Richardson ($D^2 \sim t^3$, $\lambda \sim \delta^{-2/3}$), ballistic ($D^2 \sim t^2$, $\lambda \sim \delta^{-1}$) and diffusive ($D^2 \sim t$, $\lambda \sim \delta^{-2}$) (Schroeder et al., 2012; Corrado et al., 2017).


To describe the small-scale surface circulation following the cluster of drifters, their motions with respect to the centre of mass of the cluster were considered and the differential kinematic properties (DKPs) of the surface currents were calculated. The DKPs of a flow describe how the surface water can decrease/increase in area, rotate, can be stretched or sheared (Okubo, 1970; Okubo and Ebbesmeyer, 1976; Molinari and Kirwan, 1975). They are defined by a Taylor expansion of the velocity field:


$$u = (\delta + \sigma_n)/2\ x + (\sigma_s - \zeta)/2\ y\ , \tag{3}$$

$$v = (\sigma_s + \zeta)/2\ x + (\delta - \sigma_n)/2\ y\ , \tag{4}$$


with the following DKPs, divergence: $\delta = \partial u/\partial x + \partial v/\partial y$, vorticity: $\zeta = \partial v/\partial x - \partial u/\partial y$, shearing deformation rate: $\sigma_s = \partial v/\partial x + \partial u/\partial y$ and the stretching deformation rate: $\sigma_n = \partial u/\partial x - \partial v/\partial y$, where $u$ and $v$ are the zonal and meridional velocity components, $x$ and $y$ are the zonal and meridional coordinates, respectively, in the system of reference moving with the centre of mass of the cluster.

The strain ($\rho = [\sigma_s^2 + \sigma_n^2]^{1/2}$), Okubo-Weiss parameter (OW $= \rho^2 - \zeta^2$) and instantaneous rate of separation (IROS $= \delta + \rho$) were also estimated. The OW measures the relative importance of strain and vorticity: elliptic regions (OW<0) are dominated by rotation, whereas hyperbolic regions (OW>0) are dominated by strain and deformation (Provenzale, 1999; D'Ovidio et al., 2009). The IROS is the zero order Lagrangian rate of separation at the initial time (Schaeffer et al., 2017; Lorente et al., 2021) and is therefore related to the dispersion statistics defined above, in particular to the initial exponential spreading.



There are two approaches to estimating the DKPs of horizontal currents. In the first method, small clusters of $n$ drifters (with $n >= 3$) are used to solve equations (3) and (4) using least squares (Molinari and Kirwan, 1975; Essink et al., 2019; Tarry et al., 2021). In the second method, the drifter velocities are interpolated on a uniform regular grid to directly calculate the horizontal derivatives of velocities and the DKPs (Lodise et al., 2020). In this work, we chose the second method and used the Data Interpolating Variational Analysis (DIVA, Troupin et al., 2012) to interpolate drifter velocities on a regular horizontal






grid with a cell size of 0.1 km, a signal-to-noise ratio of 1, and zonal and meridional correlation scales of 1 km. This particular interpolation method was preferred because it provides a better estimate of the

error field. In practice, interpolated values were not considered if the relative error exceeded 50%. Gradients were estimated by central finite differences of the interpolated velocity field.

## 3. Results

### 3.1 Drifter trajectories and qualitative description of the circulation

The surface drifters were released near the southern edge of a filament of coastal water extending tens of

kilometres offshore. Satellite imagery (Figure 3) shows the development and morphology of the filament whose extremities are forming a "mushroom-like" feature with anticyclonic and cyclonic eddies expanding to the North and South, respectively. The bulk of the drifters ended up in the southern cyclonic eddy. After an initial mean southward converging drift until 9 October 00:00 UTC, they turned eastward and then northward as they diverged (Figure 4).

If we focus on the initial motion of the drifters, numerous surface instruments (i.e., the CODE, CARTHE and PARC drifters deployed in the central and northeastern portions of the array) moved anticyclonically with the inertial (17.55 h) or diurnal (24 h) period (Figure 5). In contrast, SVP drifters deployed at the same locations moved directly southward, with no anticyclonic rotation. Considering the short record duration, it is not easy to separate inertial and diurnal currents. It is doubted that diurnal tidal currents

are dominant in the SLS (Poulain et al., 2018). Therefore, we can speculate that the anticyclonic rotation in the tracks is the remnant of near-inertial surface currents, which were likely generated by the storm a day earlier.

The vertical structure of thermohaline properties and currents in the area sampled by the drifters was measured by an Arvor-C profiling float and two RIVER drifters (see their initial tracks in Figure 5).

Significant shear of horizontal currents between the surface and 20 m depth was measured by the ADCP on the two RIVER drifters. The difference between the speeds at the surface and at 15m can be up to 10 cm/s (Poulain, 2020) and is compatible with the different motions of the CODE and SVP drifters discussed above. Temperature and salinity values measured by the Arvor-C float near the drifters (not shown) indicate a well-mixed surface layer with a temperature of about 21 °C and a salinity between

37.94 and 38.00 PSU, extending down to a depth of about 40 m. At this depth, there is a sharp thermocline and a minimum salinity (Poulain, 2020). Large vertical and temporal variations of salinity associated with the offshore-flowing filament were not observed, probably because the float deployed in the centre of the drifter array (Figure 5) remained outside of the filament (see satellite image in Figure 3).

### 3.2 Relative surface dispersion

All CODE, CARTHE and PARC drifters were considered together to search for pairs near the time of deployment (8 October 12:00 UTC) with selected separation distances of 100, 500, 1000, 2500 and 5000 m. Since the data set is limited to 10 October at 07:00 UTC and some drifters stopped transmitting before that time, the number of pairs may decrease with time. The initial (maximum) number of pairs and the





minimum number of pairs (at 07:00 UTC on 10 October) are listed in Table 1. They vary between 8 and
220    76.

The MSSD versus time, starting at 12:00 UTC on 8 October, is shown in a log-log diagram in Figure 6.
Despite the relatively small number of pairs, the rate of change of the MSSD with time (also called
relative diffusivity) during the first ten hours of drift appears to be significantly larger with the short
initial spacing of 100 m, than with the other larger initial distances. It can possibly be approximated by
exponential growth. For longer times, the MSSD can be approximated by a power law, with the slope
decreasing slope with increasing initial distance. However, comparison with theoretical dispersion
regimes of geophysical turbulence is not straightforward.

The FSLE for initial pair spacing between 100 m and 7 km was estimated in a similar manner, i.e., for
pairs starting on 8 October at 12:00 UTC and tracked until 10 October at 07:00 UTC. The FSLE was
calculated for scales divided into non-overlapping 100 m intervals. The results are shown in Figure 7 in
a log-log plot, along with the number of pairs in the scale intervals. Because pairs are tracked over a
limited period of 43 h some of them do not have time to separate by the prescribed amplification factor
(1.2) and do not contribute to the estimate of the average separation time, $\tau$, of equation (2).  The number
of pairs whose separation increases by 120% in less than 43 h is also shown in Figure 7. It is generally
smaller than the initial number of pairs considered, especially for large scales. The average time for the
120%, increase in separation distance varies from 0.7 h (for small scales) to 27 h (for large scales). In
general, the FSLE decreases with scale. At small scale (100-200 m) it is large (~ 5 $d^{-1}$) and fairly constant.
As scales increase (200-400 m), there is a strong negative slope. At larger separation distances (1-6 km),
the FSLE is weakly decreasing with scale, with values near 0.3-0.5 $d^{-1}$. Again, comparison with
theoretical relative dispersion slopes is not obvious.

For several reasons, the confidence intervals for MSSD and FSLE can be quite large and are not easy to
estimate. First, the number of pairs is small for small separation. Second, the distributions of squared
separation distances and separation times are generally not Gaussian and their mean values may be
meaningless. Third, the drift period of 43 h is short, and the FSLE may be underestimated because a
substantial fraction of pairs does not reach the 120% amplification factor during the tracking period.
Nonetheless, our relative dispersion results provide some useful information, as discussed later.

### 3.3 Surface DKPs

The position and velocities of the centre of mass of the cluster of CODE, CARTHE and PARC drifters
were removed from the individual positions and velocities to calculate the DKPs. Figure 8 shows the
track of the centre of mass, its velocity components, and its speed versus time. The number of drifters in
the cluster is also shown. It varies between 90 and 96. Two maxima of the speed of the centre of mass
(~20 cm/s) occurred on 8 October around 23:00 UTC and on 9 October around 22:00 UTC. The initial
slight anticyclonic rotation in the track of the centre of mass is likely due to remnant near-inertial motion





as discussed earlier. Distances between the drifters and the centre of mass were converted to kilometres to produce DKP maps.

An example of residual surface circulation interpolated by DIVA is shown in Figure 9 for 8 October at 14:00 UTC. Grid points with a relative interpolation error larger than 50% are excluded. The maximum relative speed of the drifters, in the reference frame moving with the centre of mass of the cluster, is ~17

cm/s. At this time, the mean flow is directed southeastward, with a zonal (meridional) component of 6.7 (-2.4) cm/s (Figure 8). The horizontal divergence varies between -0.7 $f$ and 0.7 $f$, where $f$ is the local inertial frequency. The centre and southeastern portion of the cluster are divergent, while convergent zones prevail elsewhere.

We now examine the DKP maps at selected times to characterize the flow within the cluster and to

monitor the shape (extent and deformation) of the area covered by the drifters. The times when some of the CODE drifters were not transmitting are skipped. Immediately after the deployments (at 14:00 UTC, Figure 10) the cluster generally expands in the meridional direction, mainly due to the drifters near the southern edge moving rapidly southward and showing significant convergence and strain.

Eight hours later (Figure 11), the northern portion of the cluster has extended zonally, and its southern

edge has formed a thin branch extending southward and turning cyclonically. The divergence is generally weak. Positive vorticity prevails east of the southward-flowing limb, while strain dominates on the opposite west side.

By the morning of the next day (9 October at 06:00 UTC; Figure 12), the southern limb has extended further in a cyclonic eddy. The divergence is patchy. A large positive vorticity exceeding $f$ occurs in the

inner part of the eddy. Outside the eddy, a hint of negative vorticity is evident. Strain is significant, especially just outside the eddy.

About a day later, at 04:00 UTC on 10 October, a few hours before the recovery operations (Figure 13), some drifters moved northward and nearly closed the loop of the cyclonic eddy in its northern sector. Other drifters have diverged and the DIVA method is not suitable for interpolating the flow between

them. Nevertheless, there is still a strong signature of positive vorticity in the eddy, and significant dispersion (strain and IROS) in the central sector west of the eddy.

## 4. Discussion and conclusions

A small cluster (scale ~6 km) of numerous Lagrangian instruments (more than 100 drifters and one profiling float) was deployed in the SLS coastal area on 8 October 2020 to characterize the near-surface

submesoscale circulation and relative lateral dispersion. The instruments were tracked for about 2 days and some of them were recovered on 10 October. During this period, the drifters were trapped in an offshore-flowing filament and a small cyclonic eddy. Satellite imagery of ocean colour (near-surface chlorophyll concentration) revealed the shape of the filament extending tens of kilometres offshore in the southwestward direction and its evolution over time into a "mushroom-like" feature with small eddies

developing at its southern and northern ends (Figures 1 and 3). The speed of the near-surface currents



measured by the drifters varied between 0 and 30 cm/s. The cluster moved toward the southeast at a mean speed of 10-20 cm/s (Figure 8).

Drifter velocities were used to estimate the DKPs and the relative dispersion of the near-surface currents on scales as small as 100 m. The DKPs within the cluster exhibit significant spatial and temporal
variability, with absolute values reaching the order of magnitude of the local inertial frequency. Significant convergence was observed in the southwestward flow of the filament. A divergence of the order of $f$ may correspond to significant vertical velocities in the upper mixed layer (Essink et al., 2019; Lodise et al., 2020; Tarry et al., 2021) leading to significant 3D dispersion of near-surface tracers (contaminants, biological organisms, etc.). Unfortunately, due the small number of SVP drifters drogued
at 15 m it is not possible to estimate vertical velocity in the study area. However, an approximate estimate of divergence at 15 m depth, based on the area rate of change method (Molinari and Kirwan, 1975) applied to the sparse coverage of independent SVP drifter triplets within the size range 2-7 km and with an aspect ratio larger than 0.2 (Esposito et al., 2021), shows an average value of $0.2\,f$, which is weaker than the magnitude found at the sea surface. Vorticity dominates in the core of the cyclonic eddy, strain
prevails at the outer edge of the eddy. The Okubo-Weiss parameter shows the alternation of elliptic (OW<0) and hyperbolic (OW>0) regions.

The relative dispersion on small scales (~100-300 m) is initially exponential and related to some of the DKPs (e.g., instantaneous separation rate, strain and divergence; Figures 6 and 7). After 5-10 h, or for initial separations greater than 500 m, the MSSD and FSLE show smaller relative dispersion rates with
a slight decrease as a function of scale. The slope of the FSLE appears to be less than Richardson's -2/3 power law. This is not surprising since this theoretical law generally applies to scales larger than 10 km (Corrado et al., 2017; Lumpkin and Elipot, 2010; Bouzaiene et al., 2020) and in our study the maximum separation scale is 7 km. Similar to Schroeder et al. (2012), maximum FSLE values between 1 and 10 day$^{-1}$ for scales smaller than 300 m confirm that submesoscale dispersion is much larger in the coastal
zone than in the open Mediterranean Sea (Lacorata et al., 2001; D'Ovidio et al., 2009) and open ocean (Corrado et al., 2017; Essink et al., 2019; Lumpkin and Elipot, 2010). In general, direct comparison of our dispersion results with the slopes predicted by two-dimensional geophysical turbulence theory is not satisfactory. This is not surprising since dispersion is due to advection by deterministic velocity fields that are highly variable in time and space, and the integration time of ~ 2 days is not sufficient to consider
dispersion as a random process. Deploying more drifters, with smaller separation distances (tens of meters) and tracking them over a longer period of time (weeks) should provide more robust results, that may be more comparable to the theoretical laws.

In general, offshore transport and dispersion of coastal waters are shown to be significant at the submesoscale (< 10 km), including fast currents (up to 50 cm/s) that change rapidly (hours). Current
operational numerical models for diagnosing or predicting coastal circulation (e.g., CMEMS, see Figure 1) are not capable of simulating this variability and therefore are not yet suitable for investigating or predicting the complex coastal dynamics, in particular the advection and dispersion of tracers, such as biological constituents (e.g., chlorophyll) and contaminants. To achieve this goal, numerical models with



higher spatial and temporal resolution are needed, possibly nested in CMEMS simulations and driven by
atmospheric models with similar resolution.

### Data Availability Statement

The data used in the study is available upon request to P.M.P. The CARTHE drifter data are available at
https://doi.org/10.17882/85161.

### Author contributions

Conceptualization, P.M.P.; methodology, P.M.P., formal analysis, P.M.P.; investigation, P.M.P..;
resources, P.M.P., C.B., S.T., L.C., M.B. and M.M.; data curation, P.M.P. and M.M.; writing—original
draft preparation, P.M.P.; writing—review and editing, P.M.P.,C.B., S.T., L.C., M.B. and M.M.; funding
acquisition, P.M.P.. All authors have read and agreed to the published version of the manuscript.

### Competing interests

The authors declare that they have no conflict of interest.

### Acknowledgments

We thank all the people who contributed to the success of the DDR20 sea trial, including the
administrative and scientific personnel of CMRE, the captains and crew of all participating ships and the
Italian Coast Guard (Capitaneria di Porto of Livorno). Special thanks go to Marina Ampolo-Rella and
Lancelot Braasch for their efforts during the experiment, and to John Waterston for providing the PARC
drifters. This research was funded primarily by the NATO Allied Command Transformation Future
Solutions Branch. M.B. contribution was supported by the JERICO-S3 project (EU funded H2020
Programme, grant number 871153).

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



Table 1. Maximum and minimum number of drifter pairs used to compute the relative dispersion statistics between 8 October 2020 at 12:00 UTC and 10 October 2020 at 07:00 UTC, for selected separation ranges.


| Initial separation range [m] | Max # pairs | Min # pairs |
|---|---|---|
| 50-150 | 9 | 8 |
| 450-550 | 32 | 30 |
| 950-1050 | 58 | 52 |
| 2450-2550 | 76 | 72 |
| 4950-5050 | 27 | 27 |



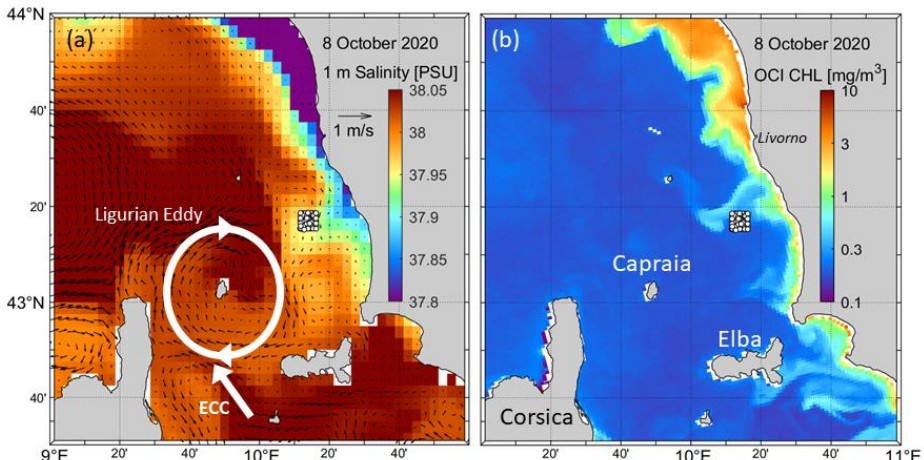

**Figure 1: (a) CMEMS near-surface currents (arrows) and salinity (colours) and (b) MODIS chlorophyll concentration (OCI algorithm) on 8 October 2020 at 12:00 UTC in the SLS. The Italian mainland is to the East. The drifter deployment locations are indicated with white dots (6x6 km² array). The ECC and Ligurian Eddy are schematized in white.**

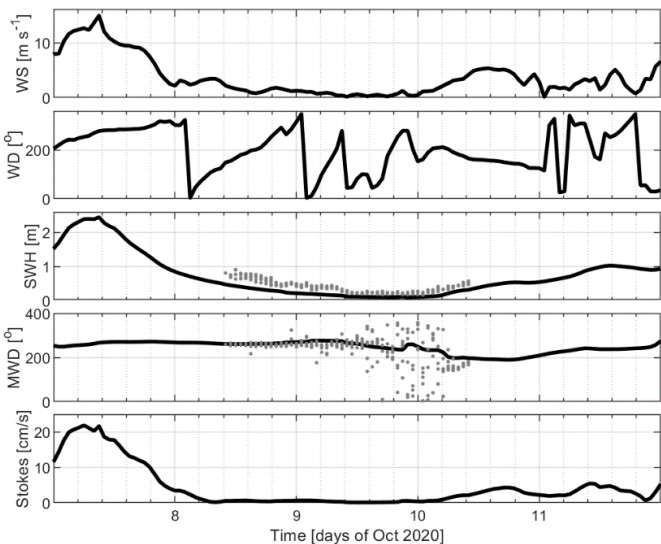

**Figure 2. ECMWF ERA5 atmospheric and surface wave products at 43°N, 10°E (black curves): 10 m wind speed (WS) and direction (WD), significant wave height (SWH), mean wave direction (MWD) and surface Stokes drift. The surface properties measured by the DWS drifters are superimposed with grey dots. Wind and wave direction are clockwise from true North (from).**



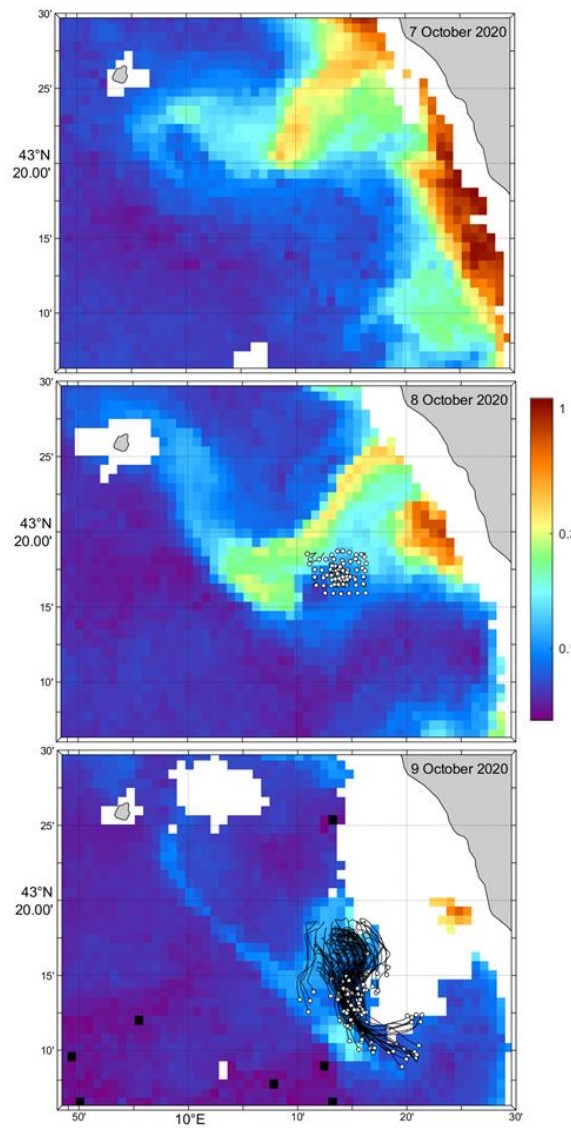


**Figure 3. MODIS chlorophyll images on 7, 8 and 9 October 2020 and tracks of the drifters from deployment until 12:00 UTC on the respective days (white circles). Chlorophyll concentration is in mg/m³.**





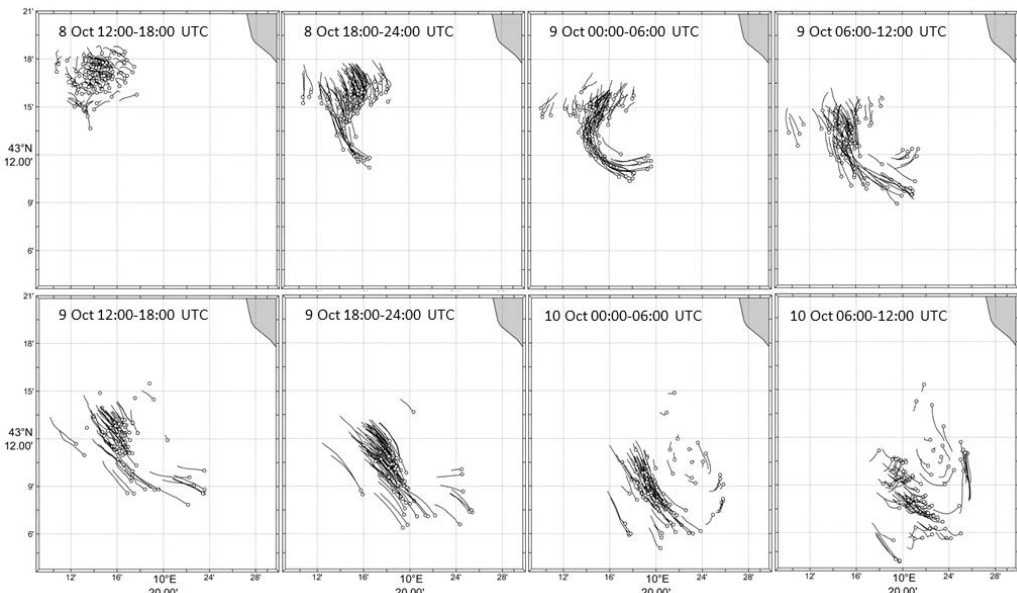

**Figure 4. Track segments of all the drifters. Segments are 6-h long and end with an open circle for each drifter on the date/time posted in the panels.**


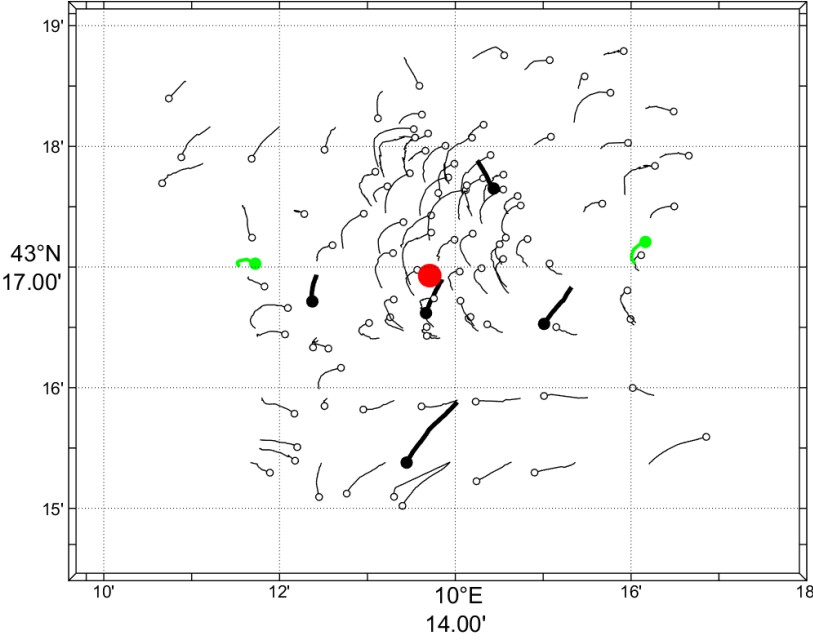

**Figure 5. Tracks between 12:00 and 15:00 UTC on 8 October for all CODE, CARTHE and PARC drifters (thin curves and open circles), for the five SVP drifters (thick curves and black dots) and for the two RIVER drifters (green). Symbols are at the end of the trajectory segments. The position of the Arvor-C float during the same period is shown with a red dot. Coherent anticyclonic motion of the surface drifters contrast with the mean southward motion of the SVP drifters.**





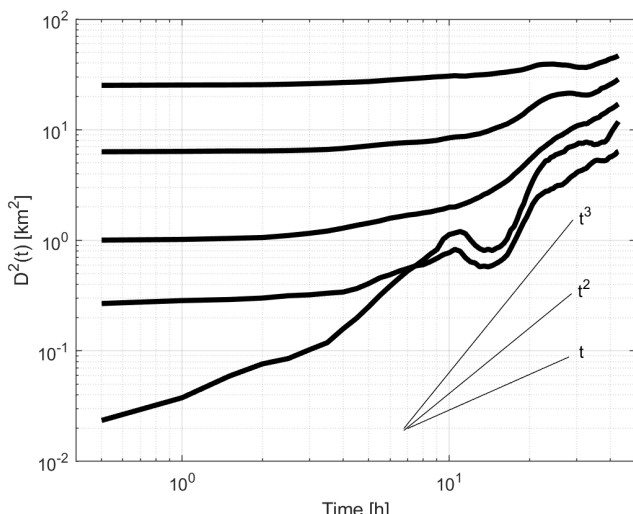


**Figure 6. MSSD versus time for selected initial distances of 100, 500, 1000, 2500 and 5000 m in a log-log plot. Initial time is 8 October at 12:00 UTC. Slope corresponding to theoretical dispersion regimes are also shown**

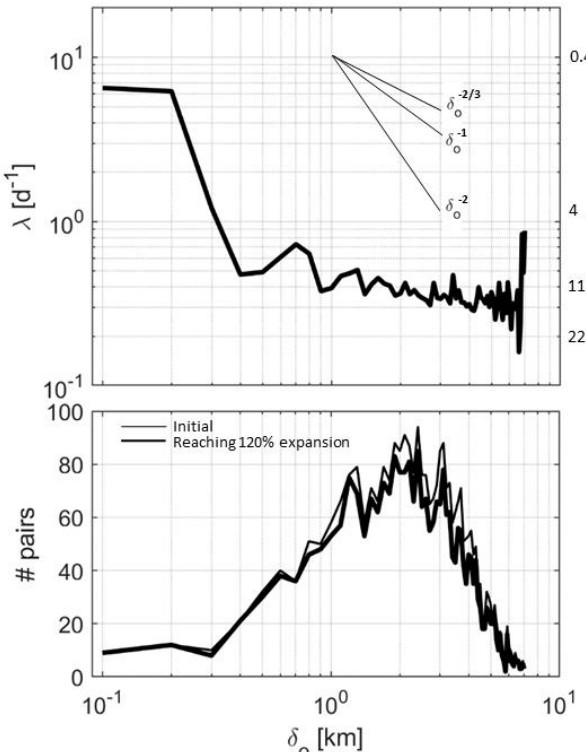

**Figure 7. Top: Scale-dependent FSLE $\lambda(\delta_o)$ as a function of scale $\delta_o$ in a log-log plot using pairs tracked from 8 October at 12:00 UTC. The diffusive ($\delta_o^{-2}$), ballistic ($\delta_o^{-1}$) and Richardson ($\delta_o^{-2/3}$) regimes are indicated by straight lines. Bottom: Number of initial pairs considered in 100 m scale bins versus scale (thin) and number of pairs whose separation distance amplified by 120% or more during the 43 h drift period (thick).**



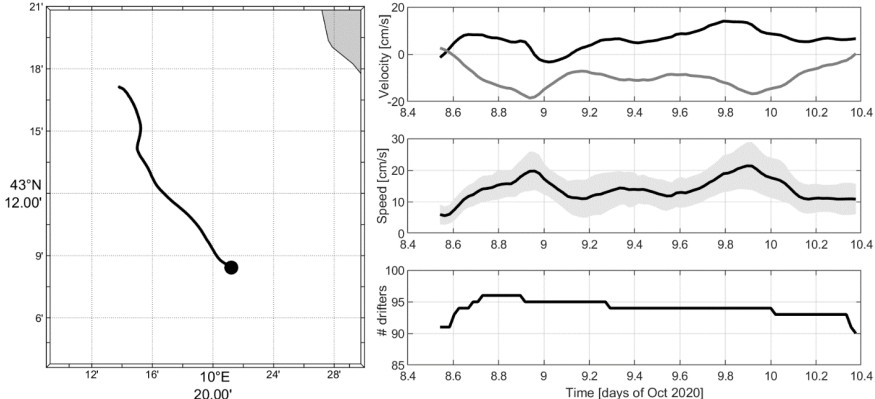


**Figure 8. Left: Track of the centre of mass of the drifter cluster between 8 October at 13:00 UTC and 10 October at 09:00 UTC (black dot). Right: Time series of the centre of mass velocity components (zonal – black, meridional – grey), speed (with shading corresponding to ± one standard deviation) and number of drifters in the cluster.**

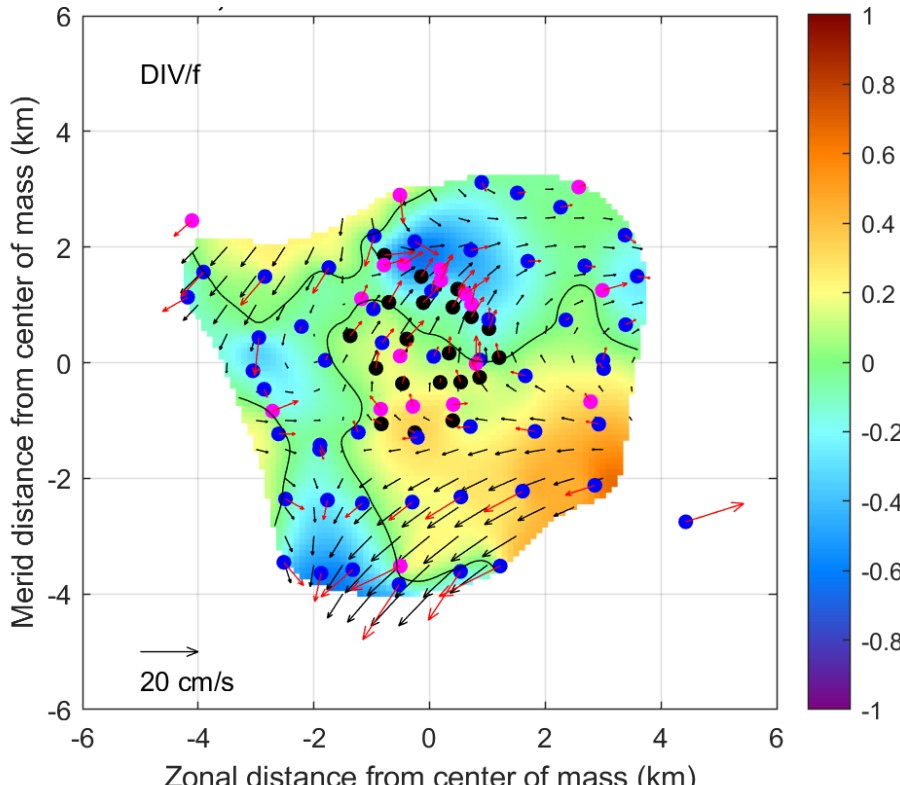


**Figure 9. Positions of the drifters with respect to the cluster's centre of mass on 8 October at 14:00 UTC (CODE – blue, CARTHE – black and PARC – magenta). The drifter relative velocities are shown with red arrows, whereas the DIVA interpolated flow is shown with black arrows. The horizontal divergence calculated with the interpolated velocity field is shown with colours. The null contour line is depicted in black.**
**Divergence is scaled by the local inertial frequency $f$.**



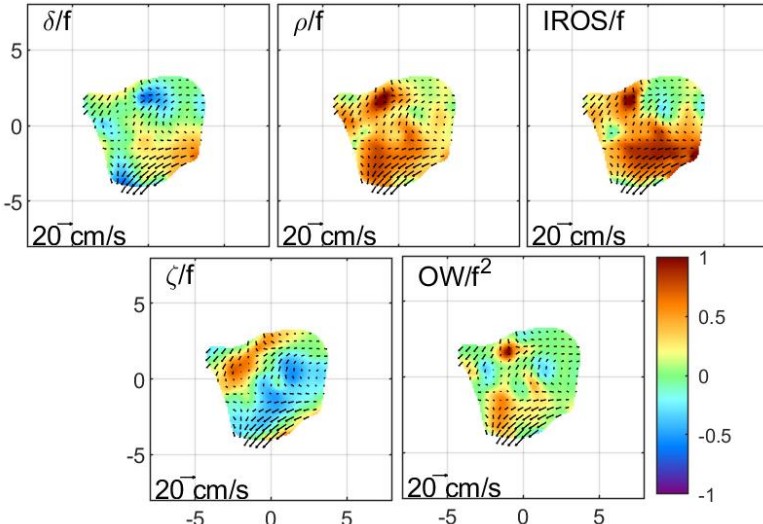

**Figure 10. Maps of relative interpolated currents superimposed with color-coded DKPs on 8 October 2020 at 14:00 UTC. See text for DKP definitions. DKPs are scaled by the local inertial frequency $f$.**


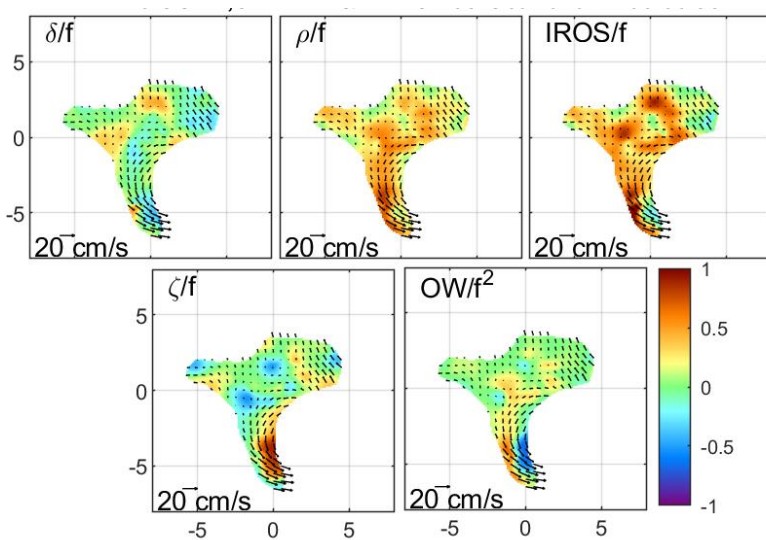

**Figure 11. Same as Figure 11 but on 8 October 2020 at 22:00 UTC.**



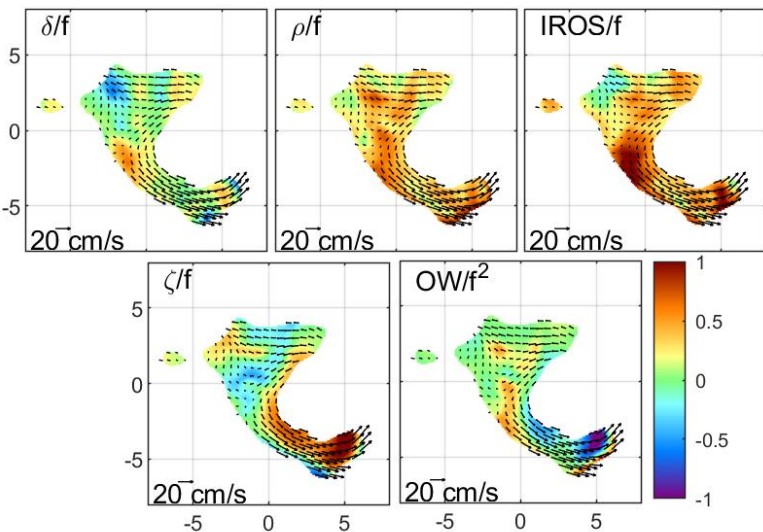

**Figure 12. Same as Figure 11 but on 9 October 2020 at 06:00 UTC.**

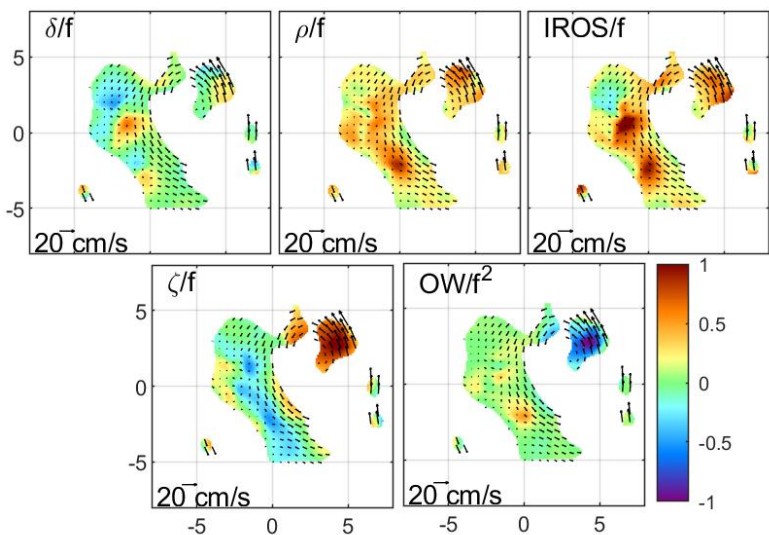

**Figure 13. Same as Figure 11 but on 10 October 2020 at 04:00 UTC**
