# Peer review of "Relative dispersion and kinematic properties of the coastal submesoscale circulation in the southeastern Ligurian Sea"

_EGUsphere, 2023_

## Author Comment (AC3)

[Figure]

**Figure 1: (a) CMEMS near-surface currents (arrows) and salinity (colours) and (b) MODIS chlorophyll concentration (OCI algorithm) on 8 October 2020 at 12:00 UTC in the SLS. The Italian mainland is to the East. The drifter deployment locations are indicated with white dots (6x6 km² array). The ECC and Ligurian Eddy are schematized in white.**

[Figure]

**Figure 2. ECMWF ERA5 atmospheric and surface wave products at 43°N, 10°E (black curves): 10 m wind speed (WS) and direction (WD), significant wave height (SWH), mean wave direction (MWD) and surface Stokes drift. The surface properties measured by the DWS drifters are superimposed with grey dots. Wind and wave direction are clockwise from true North (from).**

[Figure]

15 **Figure 3. MODIS chlorophyll images on 7, 8 and 9 October 2020 and tracks of the drifters from deployment until 12:00 UTC on the respective days (white circles). Chlorophyll concentration is in mg/m$^3$.**

[Figure]

**Figure 4. Track segments of all the drifters. Segments are 6-h long and end with an open circle for each drifter on the date/time posted in the panels. CMEMS surface currents are overlaid in gray for the central hour.**

20

[Figure]

**Figure 5. Tracks between 12:00 and 15:00 UTC on 8 October for all CODE, CARTHE and PARC drifters (thin curves and open circles), for the five SVP drifters (thick curves and black dots) and for the two RIVER drifters (green). Symbols are at the end of the trajectory segments. The position of the Arvor-C float during the same period is shown with a red dot. Coherent anticyclonic motion of the surface drifters contrast with the mean southward motion of the SVP drifters.**

25

[Figure]

30    **Figure 6. MSSD versus time for selected initial distances of 100, 500, 1000, 2500 and 5000 m in a log-log plot. Initial time is 8 October at 12:00 UTC. Slope corresponding to theoretical dispersion regimes are also shown.**

[Figure]

**Figure 7. Top: Scale-dependent FSLE $\lambda(\delta_o)$ as a function of scale $\delta_o$ in a log-log plot using pairs tracked from 8 October at 12:00 UTC. The diffusive ($\delta_o^{-2}$), ballistic ($\delta_o^{-1}$) and Richardson ($\delta_o^{-2/3}$) regimes are indicated by straight lines. Thin gray curves indicate the 95% confidence intervals. Estimate using Boffetta et al (2000)'s method (red curve). "Doubling" times are posted to the right in hours. Bottom: Number of initial pairs considered in 100 m scale bins versus scale (thin) and number of pairs whose separation distance is amplified by 120% or more during the 43 h drift period (thick).**

35

40

[Figure]

**Figure 9. Maps of relative interpolated currents superimposed with color-coded DKPs on 8 October 2020 at 12:00 UTC. See text for DKP definitions. DKPs are scaled by the local inertial frequency $f$.**

45

[Figure]

**Figure 10. Same as Figure 9 but on 8 October 2020 at 18:00 UTC.**

[Figure]

**Figure 11. Same as Figure 9 but on 9 October 2020 at 00:00 UTC.**

50

[Figure]

**Figure 12. Same as Figure 9 but on 9 October 2020 at 06:00 UTC.**

[Figure]

**Figure 13. Same as Figure 9 but on 9 October 2020 at 12:00 UTC.**

55

[Figure]

**Figure 14. Same as Figure 9 but on 9 October 2020 at 18:00 UTC.**

[Figure]

**Figure 15. Same as Figure 9 but on 10 October 2020 at 00:00 UTC.**

60

---

## Author Response (AR1)

Reviewer 1

We thank Referee #1 for his/her comments on the original manuscript.

Reviewer 2

We thank Referee #2 for his/her helpful detailed comments on our original manuscript.  We have addressed all the comments to improve the paper. Our responses to the individual comments are as follows:

A. Major drawback

1. The scientific objectives of the paper are now better stated in the introduction. The use of Eulerian and Lagrangian metrics to quantitatively describe surface dispersion is also better motivated.
2. The errors on the MSSD, FLSE and DKPs due to the drifter GPS position error is now discussed in more detail, and relevant references are added. Bootstrapping is used to provide 95% confidence intervals on the mean values involved in the MSSD and FLSE calculations. We do not believe that the agreement with theoretical slopes would be better by fitting the FLSE spectrum. For estimating the DKPs, the DIVA interpolation was used with a S/N of 10 (error of ~1 cm/s on velocities of ~10 cm/s) and decorrelation scales of 1 km.  Spatially interpolated values with relative error larger than 50% are excluded.
The errors when estimating the DKPs using the least square approach are not discussed here since this method is not used.

B. Specific comments

1. The Boffetta et al. (2000) has been added and the definition of the FSLE has been expanded.

2. We considered the average growth time rate.

3. Indeed equations 3 and 4 correspond to the first order Taylor expansion. No mean flow is added because, as stated in the text, we consider the flow with respect to the center of mass of the drifters.

4. The CMEMS currents have been added to all panels of Figure 4.  Text on the qualitative comparison with the drifter velocities has been added. We have checked that FSLE from CMEMS (actually from AVISO) are too large scale and are not relevant for this study.

5. Each curve has now a different color and the corresponding discussion has been expanded.

6. 95% confidence intervals based on bootstrapping have been added to the FSLE curve. In addition, the estimate using Boffetta et al (2000)'s method has been overlaid for comparison.

7. Figure 8 has been removed because its information content was poor.

8. Figure 9 has been removed because its information content was poor. The figures with the DKPs have been modified (more dates and better graphics). Color scaling was not changed, in particular

the zero values have been kept in green to contrast them with the white areas with no data (outside the 50% S/N level).

9. The use of mixed Eulerian (DKPs) and Lagrangian (MSSD, FLSE) metrics (or statistics) to describe the circulation and dispersion is now better motivated.

10. Indeed, several papers in the literature deals with the comparison, but generally at larger scales than those considered here, and with better success. This has been added in the revised text.

11. The horizontal divergence ranging between these values is a result, showing the significant spatial and temporal variations of the DKPs.

12. We do not understand this comment!

13. ok

14. Several relevant references have been added to support the revised text.